# The State-of-the-Art Functionalized Nanomaterials for Carbon Dioxide Separation Membrane

**DOI:** 10.3390/membranes12020186

**Published:** 2022-02-04

**Authors:** Kar Chun Wong, Pei Sean Goh, Ahmad Fauzi Ismail, Hooi Siang Kang, Qingjie Guo, Xiaoxia Jiang, Jingjing Ma

**Affiliations:** 1Advanced Membrane Technology Research Centre (AMTEC), School of Chemical and Energy Engineering, Faculty of Engineering, Universiti Teknologi Malaysia, Johor Bahru 81310, Malaysia; karchun@utm.my; 2Marine Technology Centre, Institute for Vehicle System & Engineering, School of Mechanical Engineering, Faculty of Engineering, Universiti Teknologi Malaysia, Johor Bahru 81310, Malaysia; kanghs@utm.my; 3State Key Laboratory of High-Efficiency Utilization of Coal and Green Chemical Engineering, Ningxia University, Yinchuan 750021, China; qingjie_guo@163.com (Q.G.); jiangxiaoxia402@126.com (X.J.); mjj_1022@163.com (J.M.); 4School of Mechanical Engineering, Ningxia University, Yinchuan 750021, China

**Keywords:** functionalization, nanomaterial, nanocomposite, membrane, gas separation

## Abstract

Nanocomposite membrane (NCM) is deemed as a practical and green separation solution which has found application in various fields, due to its potential to delivery excellent separation performance economically. NCM is enabled by nanofiller, which comes in a wide range of geometries and chemical features. Despite numerous advantages offered by nanofiller incorporation, fabrication of NCM often met processing issues arising from incompatibility between inorganic nanofiller and polymeric membrane. Contemporary, functionalization of nanofiller which modify the surface properties of inorganic material using chemical agents is a viable approach and vigorously pursued to refine NCM processing and improve the odds of obtaining a defect-free high-performance membrane. This review highlights the recent progress on nanofiller functionalization employed in the fabrication of gas-separative NCMs. Apart from the different approaches used to obtain functionalized nanofiller (FN) with good dispersion in solvent and polymer matrix, this review discusses the implication of functionalization in altering the structure and chemical properties of nanofiller which favor interaction with specific gas species. These changes eventually led to the enhancement in the gas separation efficiency of NCMs. The most frequently used chemical agents are identified for each type of gas. Finally, the future perspective of gas-separative NCMs are highlighted.

## 1. Introduction

Synthetic membrane is a specially designed barrier used to regulate selective transport of molecules or chemical species to achieve separation toward specific species. The exploration of membrane-based separated started in the 18th century but the deployment was limited by their poor performance and high cost [1,2]. It was the inception of asymmetric membrane using fabrication technique developed by Loeb and Sourirajan [3] in year 1962 that jumpstarted the adoption of membrane technology [4]. Attributed to the unique structure of the Loeb–Sourirajan membrane, which consists of a thin selective layer that is supported by a porous substrate, their membrane exhibited substantially higher permeation rate compared to the contemporary membranes. A decade later, Cadotte [5,6] developed a membrane with a structure similar to that of the Loeb–Sourirajan membrane but, instead of a single-step phase inversion technique, interfacial polymerization (IP) technique devised by Wittbecker and Morgan [7] was used to obtain a composite membrane that is commonly known as thin film composite (TFC). Unlike the conventional asymmetric membrane, the selective layer or skin of TFC is formed atop a pre-made porous membrane that usually comprises different polymer to that of the skin. This approach provides membranologists with the freedom to independently design and optimize the layers while the self-termination of IP led to ultrathin skin that is <200 nm [8,9]. Apart from IP, many techniques such as dip-coating [10], spin-coating [11], layer-by-layer [12,13], grafting [14], chemical vapor deposition [15], surface growth/crystallization [16,17] and sputter-coating [17,18] have been devised for the fabrication of TFC. A polymeric membrane with anisotropic structure continues to be the model for many commercial membranes available today.

While the development of membrane technology was initially centered around water treatment processes such as microfiltration (MF), ultrafiltration (UF) and reverse osmosis (RO), the interest on this technology soon expanded to various applications such as gas purification, energy storage, power generation, pharmaceutic and health care [19,20,21,22,23]. The preference towards membrane technology is largely driven by the simplicity of membrane processes which involve few moving parts and have low labor and maintenance requirements [24,25]. A membrane can deliver competitive separation output while consuming less energy and is more environmentally friendly than many conventional thermally and chemically driven separation processes [26,27]. The modularity of a membrane unit also enables the technology to be flexibly scaled and installed/retrofitted into the existing facilities, which encourages its usage for system upgrading without exhausting the precious space available in a plant [28,29]. The growing concern on global warming and energy scarcity issues in recent years has spurred the development of membrane technology as the demand for sustainable, energy efficient and environmentally friendly solutions surged [30,31,32].

The application of a membrane for gas separation is closely associated with the energy sector, in which the removal of contaminant gases such carbon dioxide (CO_2_) and hydrogen sulfide (H_2_S) is critical for improving the calorific values of the gaseous feedstock to maximize power generation [33,34,35]. Since CO_2_ and H_2_S are corrosive, the elimination of both gases can also prevent damage to system instruments and pipelines [36,37]. Contemporarily, CO_2_ separation via a polymeric-based membrane is widely explored in applications such as natural gas sweetening [30,38,39], hydrogen purification [40,41] and biogas upgrading [31,42,43]. Apart from that, membrane technology can be used to regulate the composition of a feed stream. This is particularly useful to adjust the ratio of hydrogen (H_2_) to carbon monoxide (CO) in syngas, to suit the requirements of different end uses [27,44]. Oxygen (O_2_) enrichment is another area where membrane technology has demonstrated its potential use in the industry [29,45]. Enriched oxygen can be used in oxyfuel combustion, which leads to the complete burning of fuel and the reduction in the emission of poisonous nitrogen oxides (NOx) [46,47]. Additionally, oxygen-rich air is important in many medical and chemical processing applications [48,49,50].

Membrane development is progressing at a rapid pace, and in the current limelight is a relatively new class of membrane known as the nanocomposite membrane (NCM). NCM is prepared by dispersing nano-sized material into the polymeric network of a membrane. The logic to this combination is that the membrane can benefit from the inherent advantages of both materials, i.e., (i) the superior separation property of inorganic nanomaterials [26,51,52], and (ii) the ease of processing polymeric materials that is widely available at relatively low cost [53,54]. Moreover, the incorporation of nanomaterials, also referred to as nanofiller, can boost the performance of a membrane beyond the selectivity-permeability limit [55,56]. Essentially, the introduction of NCM can drive down the cost-to-performance ratio of membrane technology, which makes it highly competitive to existing separation technologies. The mixed matrix membrane (MMM) and thin film nanocomposite (TFN) are two major schemes of NCM. MMM is the product of incorporating nanofiller into the polymeric network of asymmetric or dense membrane, while TFN is obtained by incorporating the nanofiller into either the skin, support or both layers of TFC.

There are four geometrical classifications of nanomaterials, including zero-dimensional (0D), one-dimensional (1D), two-dimensional (2D) and three-dimensional (3D), depending on how many dimensions of the material are within the nanoscale or <100 nm [26,57,58,59]. From enhancing physiochemical properties to fine-tuning separation performance, the unique alterations that can be imparted by each type of nanofillers on the characteristics of the resulting NCM are the subjects of fascination by many researchers, and have been broadly reviewed over the past decade [26,58,60,61]. An important aspect in NCM fabrication is the compatibility of nanofiller with the dispersion medium and the polymer matrix. Since most of the nanofillers are inorganics, they are in a completely different phase to that of polymer. This mismatch in properties often results in the formation of defects which causes mechanical failure and even loss of separation efficiency to the NCM. The introduction of organic fragments onto inorganic nanofillers, which can be achieved via functionalization, is the most widely adopted approach to compatibilizing the nanofillers. Most of the past reviews have focused on streamlining information about nanomaterial synthesis techniques, processing methods and applications. In the field of NCM for gas separation, topics that have been reviewed include viability of nanocomposite for carbon capture, oil and gas and biogas refining applications [62,63,64], the role of dimensionally different nanofillers [26,65] and the function of specific nanomaterial especially silica [66], clay [67], zeolite [68,69], carbon-based nanomaterial [25,70,71,72] and metal organic framework (MOF) [11,56]. To date, CO_2_-separative NCM are the most comprehensively discussed material, where primary focuses have been placed on assessing the potentials of different nanomaterials [61,73], screening of nanofiller [74] and investigating the influences of nanofillers on membrane formation [75]. Another important aspect in the development of NCM is the modification or functionalization of nanofillers, which has been reviewed in terms of the role of modification techniques and chemical agents to stabilize nanofiller dispersion and compatibilize nanofiller with a polymer matrix. However, these reviews are made in a generic term [76] or mainly focused on a specific type of nanofiller [60,77]. Discussion on nanofiller functionalization employed in the development of gas separative NCM is fairly limited [78] and emphasized on CO_2_ separation [79,80]. In this review, the nanofiller functionalization using various types of chemical agents to fine-tune gas separation performance of NCM is discussed in depth. Other than enhancing nanomaterial homogeneity and compatibility with a membrane matrix, the roles of functional groups for tuning the pore structure and chemistry of nanofiller to achieve specific purposes were also discussed. The common analytical techniques used to validate the effectiveness of nanofiller functionalization were presented and the important chemical groups used to target CO_2_ and H_2_ separation were identified for reference for future works. From empowering defect-free NCM formation to altering synthesis mechanisms which breathe new life into the geometry and functionality of nanomaterials, surface functionalization of nanomaterial is essential for advancing NCM for gas separation. This review focuses on the shift in technological landscape of a gas-separative membrane brought by nanofiller functionalization in the past 5 years.

## 2. The Roles of Nanofiller in Gas Separation Membrane

The geometry and chemical characteristics of incorporated nanofillers can have different implications on the physiochemical properties of the host polymeric network [61,81]. Commonly, incorporation of nanofillers could increase the fraction-free volume (bn) of the membrane through the creation of voids that are associated with the disruption of the membrane polymeric network by the nanofillers [11,82]. Through X-ray diffraction (XRD) analysis, Kardani et al. [83] observed a weakened characteristic crystalline peak of polyamide (PA) segment in poly (ether-block-amide) (PEBAX) when a copper-zinc bimetallic imidazolate framework (CuZnIF) was incorporated into the membrane matrix. This was attributed to the disruption in chain-to-chain interactions by the nanomaterial, which diminished the chain packing and led to a more amorphous PEBAX matrix. Similar changes in membrane crystallinity have been reported by Azizi et al. [84] and Mirzaei et al. [85], who incorporated zinc oxide (ZnO) in a PEBAX matrix and a palladium-filled zeolitic imidazolate framework (Pd@ZIF-8) in Matrimid, respectively. The reduction in membrane crystallinity could lower the activation energy required by the penetrant molecules to diffuse from one vacant spot to another while moving across the polymeric network [86,87,88]. Ultimately, as the resistance towards diffusion is lowered, the overall gas permeability of the membrane is enhanced [89,90].

While changes in chain packing and crystallinity are the dominant effects when nanofillers are incorporated into dense membrane or selective layer, nanomaterials in porous membrane have different implications on the membrane properties. Based on Brunauer–Emmett–Teller (BET) analysis, Khdary and co-worker [91] observed a decrease in membrane pore size from 10.8 nm to 6.3 nm when 20 wt.% of N-(3-(trimethoxysilyl propyl) ethylenediamine) functionalized silica (PEDA-SiO_2_) was incorporated into a porous poly(vinylidene fluoride-hexafluoropropylene) (PVDF-HFP, Figure 1a) network. This was ascribed to the deposition of nanofiller on the walls of membrane pores, which resulted in partial pore blockage as shown in Figure 1b. Concurrently, ascribed to the high surface area of nanofiller, the specific surface area (SSA) of the composite membrane rose from 3.8 m^2^·g^−1^ to 116.4 m^2^·g^−1^, which was accompanied by an increase in CO_2_ sorption capacity from 1.7 mg·g^−1^ to 26.3 mg·g^−1^.

The enhancement in membrane SSA is more prominent when highly porous nanofillers are employed. The abundantly available nano-sized pores on the nanofillers could render large active sites for the adsorption of specific type of gas [95,96]. Since the aperture of the nanofiller pores can be precisely tuned and has a narrow pore distribution, high specificity towards interested/targeted gas molecule can be achieved [97,98]. Moreover, the pores of penetrable nanofillers such as carbon nanotube (CNT), titania nanotube (TNT), halloysite nanotube (HNT) and MOF can function as nanochannels that preferentially allow molecules that are smaller than the pores aperture to rapidly diffuse through the nanofiller [83,99]. Other than their pore system, nanofillers can also serve as templates for creating nanometric voids or channels within the polymer matrix [100,101,102,103]. This is commonly achieved by removing the incorporated nanomaterials from the membrane matrix after the formation of the nanocomposites. Lee et al. [104] employed water-soluble MOF as a template for the fabrication of porous membrane where the porosity of the membrane was increased from 73% for neat polyacrylonitrile (PAN) to 84% upon the removal of the nanofillers. Recently, Jackson et al. [94] showed the viability of producing ultrathin polymeric layer with vertically continuous nanopores using nanoparticles templating technique. They first created a monolayer of Fe_3_O_4_ by assembling the nanoparticles from the suspension containing alkanethiols, which functioned as organic ligands. Next, the organic ligands coated around the nanoparticles were crosslinked using electron beam irradiation to obtain a nanocomposite layer (Figure 1c-i) followed by chemical etching to remove the Fe_3_O_4_ template (Figure 1c-ii). The authors also controlled the thickness and 3D structure of the porous layer (Figure 1c-iii) by manipulating the parameters of irradiation.

On the other hand, the inclusion of non-permeable nanofillers such as metal oxide [105], silica [106], clay [107], graphene [108,109] and MOF-based nanosheet [110,111] can increase the tortuosity of membrane matric which divert the flow of gases through a winding path. This significantly affects the permeation rate of gases, especially gas molecules with larger kinetic diameters, as they experience greater resistance from the polymeric network during their travel through the elongated path. Among the non-permeable nanofillers, 2D nanosheets have gained tremendous attention in the development of high-performance NCMs [112,113,114]. Fascination on this type of nanofillers is driven by their unique geometry which enables the construction of various 3D assemblages [115,116,117]. The nano-range thickness permits the fabrication of defect-free layers under 1µm thick as shown in Figure 1d,e [118,119,120,121]. Considering that the thickness of selective layers is directly relatable to the resistance on molecular transport across the membrane, the ability to fabricate ultrathin separative layer is enticing for the development of highly permeable membrane.

Apart from the potential to improve separation performance, attributed to the rigidity and superior physical properties of inorganic materials, the incorporation of nanofillers can enhance the mechanical strength and thermal endurance of the resultant NCM [122,123]. Wang et al. [124] improved the tensile strength of their membrane 3.8-fold through the blending of 1 wt.% Am-BN into the polymer matrix, while Kausar [125] reported a 14% increase in the degradation temperature of the NCM when 3 wt.% of oxidized graphene nanoribbon was incorporated. Sutrisna et al. [126] inferred that Universitetet I Oslo or University of Oslo MOF (UiO-66) nanoparticles could interact with the PA segment in PEBAX via hydrogen bonds. This restricted the mobility of PEBAX chains and rigidified the membrane, which led to increase in melting point and crystallinity from 160 °C to 164 °C and 18% to 21%, respectively. Moreover, as shown in Figure 2a,b, the resulting NCMs were also more resistive to compaction as, during decompression, they could reproduce similar separation performance exhibited in the compression stage, while the neat PEBAX suffered from performance loss after compression due to irreversible changes in membrane structure. Jahan et al. [127] noted that cellulose nanocrystal (CNC) incorporated polyvinyl alcohol (PVA) could exhibit lower water uptake than that of neat counterpart, as the formation hydrogen bonds between CNC and PVA could reinforced the connectivity of polymeric chains. This led to a polymeric network that is more rigid and resistive to swelling. As can be seen from Figure 2c,d, the resulting nanocomposites were able to retain high tensile strength and elastic modulus, while the neat PVA rapidly lost its strength when exposed to highly humid environment (relative humidity of 93%). Clearly, the addition of inorganic materials, regardless of their shape, has profound impact on the physical properties of the resulting nanocomposite. Since the gaseous streams from industrial processes may operate at an elevated temperature in the range of 40–900 °C [128,129,130,131] and with pressure up to 35 bar [132,133], NCM with enhanced mechanical and thermal properties are advantageous as they can endure harsh operating conditions compared to a conventional polymeric membrane. This in turn helps to reduce operation costs for cooling and/or recompression of gaseous streams.

Aging and CO_2_-induced plasticization are two prominent problems which adversely impact the performance of gas separative membranes. Aging is a process whereby the polymeric chains of membrane rearrange into a more relaxed network to relinquish from their non-equilibrium state [134,135,136]. As a membrane ages, its porous structures collapse and led to diminished FFV, which contributes to a loss of permeability. On the contrary, CO_2_-induced plasticization is a pressure-dependent phenomenon described by the swelling of a polymeric network by absorbed CO_2_ [33,137]. This led to increased polymeric chain mobility, which has detrimental effects on the sieving capability of membrane. Ultimately, the plasticized membrane will suffer loss of selectivity. Owing to the enhanced physical properties, NCMs are often more resilient to aging and CO_2_-induced plasticization [135,138,139,140]. The improvement is closely related to the immobilization of polymer chains, which reduces the tendency of swelling or relaxation of the polymeric network [127,141,142,143].

In the previous discussion, the insertion of nanofiller into a membrane matrix was associated with the disruption of polymeric chain arrangement, which led to low membrane crystallinity. However, many studies have also found that the opposite effect could take place, depending on the strength of filler–polymer interaction and filler dispersity [144,145]. Using Fourier transform infrared (FTIR) spectroscopy, Zhang et al. [146] noted that HNT could interact strongly with PEBAX via hydrogen bonding, which limits the mobility of the polymeric network and led to increased membrane crystallinity. Similarly, Liu et al. [82] reported the formation of hydrogen bonds between nickel-based MOF (MOF-74-Ni) and polymer of intrinsic microporosity (PIM) which has limited the rearrangement of the PIM matrix. The presence of 10 wt.% MOF-74-Ni was able to slow down the aging of PIM, which was observable through the lower degree of reduction in CO_2_ permeance at 35%, compared to 68% that of pristine membrane after exposure to air for 7 days under ambient condition. Apart from resistance to permeability loss over time, Hou et al. [135] monitored the aging of PIM via solid-state nuclear magnetic resonance (ssNMR) spectroscopy. They suggested that the rate of loss in resonance intensity of ^13^C upon pulse excitation (T_1_) is inversely correlated to the relative mobility of atoms, i.e., higher T_1_ indicated greater rigidity of polymeric network and vice versa. On the other hand, Xiang et al. [56] reduced the loss in CO_2_/CH_4_ selectivity of crosslinked poly(ethylene oxide) (XLPEO) membrane from 40% to 9% by incorporating 30 wt.% chemically modified ZIF-7. The improved plasticization resistance was ascribed to the formation of chelates between the metal nodes in ZIF-7 and the ester moieties in XLPEO, which elevated the rigidity of XLPEO network. All these studies clearly exemplified the function of embedded nanofiller in altering the structural properties of polymeric material to achieve desirable separation properties. In efforts to extend the service life of modern membranes that are progressively thinner, the use of nanofillers has become indispensable. This is particularly true for ultrathin membranes as they are susceptible to aging and plasticization [2,82,147,148].

It is important to note that the extent of alteration in membrane properties caused by nanofiller incorporation depends greatly on the extent to which the inorganic materials are assimilated into the membrane matrix, as they have different phases. Interaction between nanofiller and polymer is the key to bring out the synergistical benefits of nanocomposite materials [149]. Swain et al. [150] studied the effects of incorporating silica-graphene oxide (SGO) hybrid on the mechanical properties of polysulfone (PSF) and demonstrated the importance of surface functional groups on the filler–polymer interaction. By depleting the oxygenated groups available on the SGO hybrid filler via the reduction process, the authors noticed a drop in the degree of enhancement in membrane tensile strength, as a result of the incorporation of SGO, from 147.6% to 128.6%. The authors attributed this change to the inferior dispersibility of reduced SGO. Its poor dispersibility led to agglomeration of reduced SGO, which prevented homogeneous blending of the nanofiller within the PSF matrix. Ultimately, the formation of regions with poor filler–polymer interaction inevitably resulted in a defective membrane. In most studies, the changes in the membrane’s physical and thermal properties after nanofiller incorporation has been used as an indicator to assess the effectiveness of filler–polymer interaction [123,151,152,153,154]. Apart from improving the compatibility of nanofiller with the polymer matrix, the functional groups can enhance the affinity of the membrane towards certain gas molecules via chemical interactions such as acid-base reaction, polar attraction, and hydrogen bonding [83,97]. This improves the solubility of gases into the membrane matrix, thereby elevating their permselectivity. Considering the important role played by the functional groups attached to the nanofillers to promote the formation of defect-free and high-performance NCM, it is unsurprising that many works have been devoted to the surface modification of inorganic nanofiller via functionalization [90,93,155,156].

## 3. Functionalization of Nanofiller

The high SSA of nanofillers is a double-edged sword. On the one hand, a large quantity of gas adsorptive sites is available to enhance separation performance, but on the other, the great Van de Waals forces generated by the large surface of contact led to self-aggregation of the nanofillers [81,157,158]. This issue is especially prominent with fibrous or tubular nanofillers such as CNT [159,160], carbon nanofiber (CNF) [161] and CNC [162], due to their high aspect ratio and tendency to intertwine among themselves. To this end, functionalization of nanofiller is most straightforward and effective approach to overcome the aggregation issues. Functionalization is an approach that introduces new chemical agents in/onto the nanofiller to attain certain characteristics that are unavailable in pristine nanofiller and are currently heavily adopted in various fields. Through functionalization, new functionalities, features or properties could be introduced to a material by altering its surface chemistry, which can be achieved covalently or non-covalently. Covalent functionalization involves bonding the functional groups permanently to the structure of nanofiller, while non-covalent functionalization utilizes non-permanent bonding such as Van de Waals, electrostatic and hydrogen bond interactions to attach chemical agents onto the surface of nanofiller. Both approaches have their own advantages and limitations, with the former being described as destructive but providing stable and long-lasting functionality to the nanofiller, while the mild nature of non-covalent functionalization can preserve the intrinsic properties of nanofiller, but the functional group may detach when subjected to external forces or changes in environmental conditions. Nonetheless, the introduction of functional groups can aid with the dispersion of nanofillers through steric hinderance and charge repulsion [163,164]. Furthermore, insertion of functional groups into the gallery of multilayered 2D nanomaterials can promote exfoliation and prevent restacking of nanosheets [165]. Segregating individual nanofiller from its bulk via top-down approaches exposes the active sites and pores of nanofiller that are otherwise inaccessible [166,167]. The contribution of functionalization is far more diverse than just improving nanofiller dispersity. It can be employed to alter the structure of nanofiller and fine-tune the chemical characteristic of nanofiller to enhance interaction with a polymer matrix as well as elevate gas-specific affinity. These changes could impose substantial implications on the gas separation performance of NCM, and the effects are discussed in the following sub-sections.

### 3.1. Compatibilizing Nanofiller with Solvent and Polymer Matrix

Typically, nanofiller is the first component to be dispersed in solvent prior to the addition of a polymer or monomer during the preparation of membrane precursor solution. Poor compatibility of nanofillers with solvent will result in phase separation which led to the formation of nanocomposite with two distinct layers, i.e., nanofiller-lean and nanofiller-rich phases. This inhomogeneity is highly undesirable as the properties such as thermal expansion and elasticity of both layers are extremely different, which could lead to mechanical failure and delamination of one layer from another. Apart from the goal of attaining homogeneous and defect-free NCM, maintaining the stability of nanofiller suspension is crucial to practicably process the material and avoid loss of nanomaterial through sedimentation during storage. Moreover, sedimented or aggregated nanofillers can clog the membrane fabrication machineries, leading to equipment damage and process downtime. One may ascertain the stability of nanofiller dispersion by performing the following:observing the settling rate of nanofiller in a suspension:simple sure-fire method to differentiate the relative stability of one suspension from another but time consuming, especially for very stable solutions [163,167,168,169].inspecting the suspension density and homogeneity via optical or electron microscopy:rapid visual assessment of suspension dispersibility but results may be affected by sample preparation [81,170,171].determining the surface charge of the nanofiller via zeta potential (ζ) analysis:suitable for highly charged nanomaterials and provides indicative information regarding suspension stability. Generally, nanomaterial suspension with a magnitude of ζ magnitude greater than 30 mV is considered stable, but the results may be affected by the material characteristics such as hydrophilicity, polarity, geometry and chemical groups. Hence, additional supporting characterizations are required [81,153,172,173].approximating the Hansen and/or Hildebrand solubility parameters of suspension:fairly accurate prediction of the suspension stability and provides information on dispersion mechanisms in terms of dispersive force, dipole interaction and hydrogen bonding parameters. Other molecular interactions such as electrostatic force, metallic interaction and ionic bond are not accounted for [173,174,175,176,177].

Although lowering the nanofiller loading and increasing the viscosity of polymer dope solution can be used to delay the settling duration of nanofiller, these approaches are not always applicable and limit the nanofiller usage [145]. Therefore, on-going development is rigorously carried out to overcome these drawbacks so that the reliability and processability of NCM can be improved.

To this end, functionalization of nanofiller is widely adopted due to its effectiveness and reliability [178,179,180,181]. For example, by modifying aminated UiO-66 (UiO-66-NH_2_) with palmitoyl chloride, Liu et al. [182] were able to improve the stability of the nanomaterial suspension, which would otherwise rapidly settle in cyclohexane (Figure 3a). It was suggested that the introduction of non-polar alkyl groups from palmitoyl chloride onto UiO-66-NH_2_ compatibilized the nanomaterial with the non-polar cyclohexane. In another study by Yoo et al. [173], the modification of graphene oxide (GO) with polyetheramine (PEA) enabled the nanomaterial in a wider range of solvents which remain stable for over 50 days compared to pristine GO (Figure 3b). The authors attributed the improvement to the surfactant-like nature and solubility of PEA. They also found that molecular dispersion force and polar interaction were the main contributors to the dispersibility of PEA-GO. Essentially, both Liu and Yoo noted that good nanofiller dispersion could be achieved by introducing chemical agents with similar characteristics to the solvents, i.e., polar agent for a polar solvent, non-polar agent for a non-polar solvent and an agent capable of forming a hydrogen bond for a protic solvent and vice versa. Furthermore, a recent study by Benko et al. [183], which probed the electronic state of CNT surfaces, concluded that the type and coverage density of functional groups introduced have a strong influence on the polarity of the functionalized nanomaterial, as shown in Figure 3c. This implied that functionalization could enable the fine-tuning of the nanomaterial surface properties.

In the past, incompatibility between the inorganic and polymeric materials has been a major hiccup in the development of NCM. The surface properties of neat inorganics are typically very different from that of organics hence results in poor adhesion between the polymers and inorganic nanofillers [32]. This causes an undesirable “sieve-in-cage” interface between nanofiller and polymer, shown in Figure 4, which could lead to the formation of a defective membrane.

Nowadays, the nanofiller–polymer interfaces of NCMs are commonly compatibilized via functionalization [185,186]. Priming is a rudimentary but effective technique to prepare homogeneous nanocomposite by wrapping the nanofiller within a dilute coating of the host polymer prior to nanofiller incorporation [89]. This approach ensures that the nanofiller is imparted with the same solubility characteristic as the host polymer, which promotes good nanofiller dispersion in dope solution and ultimately homogeneous membrane. However, like many non-covalent functionalization, in this approach, chemical agents are attached onto the nanofiller via non-permanent bonding [123,187]. Hence, the adhesion of chemical agents could be compromised by external factors such as change in pH, presence of foreign ions and fluctuation in temperature [188,189,190,191]. Alternatively, covalent functionalization avoided the issue related to detachment of chemical agents as they are permanently bonded and become part of the nanofiller surface structure. The work by Molavi et al. [145] and Katayama et al. [184] on the UiO-66-incorporated membrane evidenced the feasibility of grafting the host polymers onto the nanofillers. The covalently grafted polymers served as chemical bridges that enhance nanofiller–polymer interaction, which circumvented delamination of polymer matrix from the nanofiller surfaces.

Host polymers are not the only viable chemicals that can be employed for functionalization. Azizi et al. [84] showed that functionalizing zinc oxide with oleic acid (ZnO-OA) could alter the surface energy of nanoparticles to match that of the membrane matrix, which led to better compatibility between the two phases. The enhanced polarity of the functionalized nanofiller (FN) also improved its dispersion in dimethylformamide (DMF), which contribute to the uniform distribution of the ZnO-OA within the membrane. Likewise, Mahdavi et al. [39] compatibilized their silica (SiO_2_) with PEBAX matrix by introducing 1-butyl-3-methylimidazolium hexafluorophosphate ([Bmim][PF_6_]) onto the nanofillers. They suggested that [Bmim][PF_6_] could interact with PEBAX chains via dipole–dipole attraction, which lowered the difference in surface energy between SiO_2_ and PEBAX. Chen et al. [192] reported that tannic acid (TA) grafted on ZIF-8 could undergo crosslinking reaction with amino groups in the poly(vinylamine) (PVAm) selective layer, while complexation of TA with Fe^3+^ lead to strong interaction with polydimethylsiloxane (PDMS) intermediate layer. This study revealed that the functionalization not only enhanced embedment on nanofiller into the matrix of selective layer but also promoted good adhesion between the hydrophilic PVAm and the hydrophobic PDMS layers. Other than that, a study by Liang et al. [185] suggested that the introduction of functional group could alter the behavior of nanomaterial-mediated polymer nucleation. They noted that the addition of pristine and functionalized CNTs accelerated the crystallization of poly(lactide acid) (PLA) due to their high SSA which served as nucleation sites (Figure 5a). However, the crystallization activity driven by FNs was lower than that of pristine CNT. This is ascribed to the diminishing of the nucleation sites by the functional groups, which also sterically hindered the attachment of polymer chain onto the nanofiller. Consequently, nucleation rate of polymer was reduced and led to low nucleus density. Conversely, functional groups which could promote good nanofiller–polymer interfacial interaction are favorable to enhancing nucleation activity as the polymer was readily expose to the nucleation sites on CNT. In short, the incorporation of FN could affect the quality of polymeric matrix of the resulting NCM.

At this point, one might wonder how the effectiveness of functionalization was gauged. At the time of this writing, there is no systematic approach to quantify the strength of a nanofiller–polymer matrix interaction. In most studies, the cues from the difference in NCM physical properties relative to that of pristine polymeric membrane are used to justify the appropriateness of a nanofiller–polymer interaction. This was apparent from our previous discussion regarding the impact of nanofillers incorporation on the mechanical strength, crystallinity and thermal endurance of the membrane. Since this approach does not measures the strength of a nanofiller–polymer interaction directly, it is often supplemented by one or more analyses, such as the following:evaluate changes in chemical properties:commonly conducted via FTIR to detect formation of new bonds between nanofillers and polymer matrix. [192,194,195,196]visually observe nanofiller–membrane matrix interfaces:can be conducted using field emission scanning electron microscopy (FESEM) or transmission electron microscopy (TEM). This analysis focuses on a small sample area; hence, scanning of multiple random spots is recommended to obtain reliable and accurate generalization of the nanofiller–polymer interface morphologies [135,155,197,198,199]evaluate changes in membrane separation behaviorsimple indicator of presence or absence of non-selective void at the nanofiller–polymer interfaces [93,200,201,202,203,204]

Mozafari et al. [111] assessed the effectiveness of functionalization by computing the specific energy to break the bonds between nanofiller and polymer (ΔE), as shown in Equation (1):ΔE = E_c_ − (E_f_ + E_p_)(1)
where E_c_, E_f_ and E_p_ refer to the specific energies (J·g^−1^) to dissociate the nanocomposite, nanofiller and pristine polymer, respectively. Greater positive ΔE is indicative of a stronger nanofiller–polymer interaction.

Overall, functionalization is no doubt very important to the development of homogeneous nanocomposite and enables the merger of two distinctively differing materials. Yet, functionalization alone is not a silver bullet to successful formation of NCM. It is important to note that the amount of nanofiller that can be loaded into a suspension or polymeric system has a threshold such as that shown in Figure 5b. Excessive nanofiller loading can result in defective or underperforming membranes, due to agglomeration and uneven distribution of nanofillers [32,124,205,206,207,208]. Functionalization has been demonstrated to alleviate the severity of overloading-induced agglomeration. By functionalizing SiO_2_ with γ-glycidyloxypropyltrimethoxysilane (GOTMS), Ahmadizadegan et al. [209] were able to increase the loading threshold of nanofiller suspension from 10 wt.% to 15 wt.%. Beyond 15 wt.%, the negative impacts associated with nanofiller overloading would persist. Apart from that, overly modifying the nanofiller could also lead to negative consequences [81,90]. For instance, Noroozi and Bakhtiari [90] found that loading TNT with 90% tetraethylenepentamine (TEPA) resulted in lower CO_2_-adsorption capacity than that of 70% TEPA-loaded TNT. This was attributed to the blockage of nanofiller active sites or pores by the excessive functionalization agents, which leads to plugged sieve phenomenon (Figure 4). Considering the absence of pores, incorporation of plugged sieve or clogged filler into polymer matrix is analogous to the incorporation of non-permeable filler whereby tortuous diffusion paths may be created, which reduce the overall gas permeability of the resulting membrane.

### 3.2. Tuning Nanofiller Pore Size

Pore blockage of FN is not necessarily an undesirable phenomenon as it can be exploited to manipulate the sieving properties of nanofiller. Liu et al. [99] noted that even though the permeability of their nanocomposites decreased when the incorporated MOF was filled with higher ionic liquid (IL) content, the membrane selectivity was gradually improved. The filling of more 1-butyl-3-methylimidazolium bis(trifluoromethylsulfonyl)imide ([Bmim][Tf_2_N]) into MOF led to thicker deposition of the IL on the wall of a nanofiller pore. This contributed to a more constrictive pore that has greater mass transport resistance which enhances the sieving property of the nanofiller. Likewise, Mozafari et al. [111] found that substituting the original ligand of UiO-66 with a larger chemical agent reduced the MOF pore diameter from 1.74 nm to 1.69 nm. Additionally, the introduction of amine groups (-NH_2_) into the MOF structure via the substituted ligand enhanced the nanofiller affinity towards CO_2_. The contribution of these two changes improves the overall selectiveness of the corresponding NCM towards CO_2_ over CH_4_. In another work, by substituting the 1,4-benzenedicarboxylic acid (H_2_bdc) ligand of UiO-66 to 2-amino-1,4-benzenedicarboxylic acid (H_2_bdc-NH_2_) and 1,2,4,5-benzenetetracarboxylic acid (H_2_bdc-(COOH)_2_), Sutrisna et al. [126] were able to reduce the nanofiller pore diameter from 1.50 nm to 1.45 nm and 1.33 nm, respectively. Ebadi Amooghin et al. [206] reported that the replacement of the sodium ion (Na^+^) from the core of zeolite Y with cobalt ion (Co^2+^) increased the nanofiller pore diameter from 1.92 nm to 2.01 nm due to the smaller size of Co^2+^ (0.75 Å) compared to Na^+^ (1.02Å). On the other hand, complexation of Co^2+^ with 1,3-phenylenediamine followed by reaction with 2,4-pentanedione gradually fills the zeolite cavity and reduces its pore size to 1.98 nm. All these studies show that functionalization could be utilized to control the characteristics of nanofiller so as to achieve the desired separation characteristic. Nevertheless, functionalization that was carried out in situ during nanofiller synthesis could alter the growth rate, shape, size and crystallinity of nanofiller [56,210,211,212].

### 3.3. Enhancing Gas Separation Performance

Discussion on functionalization has, thus far, focused on the changes in NCM separation properties associated with sieving ability, which stems from the alteration of physical characteristics of the membrane matrix. For instance, both the reduction in nanofiller aperture by partial blockage of functional groups and the enhanced network rigidity due to improved nanofiller–polymer interaction could also lead to a membrane matrix that is more capable of sieving out larger gas molecules. Yet, sieving is a size-based separation mechanism which is not effective in differentiating gas species with similar dimension. To circumvent this limitation, chemical agents that can selectively interact or react with the targeted gas should be employed. This empowers the resulting NCM with gas-specific selectivity, a feature commonly found in CO_2_-selective membranes.

#### 3.3.1. Functional Groups with Gas-Specific Selectivity

CO_2_ is a polar acidic gas that can readily react with many Lewis bases such as amine, poly(ethylene glycol) and water [37,213,214]. Based on the data compiled in Table 1, to date, molecules containing amine-based groups are by far the most widely adopted functionalization agent used to enhance the selectiveness of nanofillers towards CO_2_ [215,216,217]. The Lewis acid-base reaction between amine (R-NH_2_) and CO_2_ that form carbamate (-NHCOO^−^), as shown in Equation (2), has been well understood [75,90].
(2)$${R-\mathrm{NH}}_{2}{+\mathrm{CO}}_{2}⇌{\;R-\mathrm{NHCOO}}^{-}{+H}^{+}$$

Attributed to this reversible reaction, amine- or amino-FNs are endowed with good CO_2_-affinity [38,218]. As discussed earlier, Sutrisna et al. [126] reported a reduction in the pore size of UiO-66 after amine-functionalization, which was accompanied by a decrease in SSA from 1800 m^2^·g^−1^ to 1213 m^2^·g^−1^. Although the active surface of the FN was diminished, its CO_2_ adsorptivity increased from 2.2 g·cm^−3^ to 2.9 g·cm^−3^. This clearly showed that the introduction of the amine group into UiO-66 has a positive implication on preferential CO_2_ sorption of the nanofiller. Noroozi et al. [90] also reported a similar observation, whereby functionalizing TNT with tetraethylene pentaamine (TEPA) improved the CO_2_ sorption capacity of their nanofiller by 368% from 0.71 mmol·g^−1^ to 3.32 mmol·g^−1^. When the FNs are incorporated into the polymeric matrix, this characteristic could be carried-over to the nanocomposite, which enhances the membrane selectiveness towards CO_2_ over other gases [32,91,219,220].

Apart from a Lewis acid-base reaction, CO_2_ could interact with polar functional groups such as hydroxyl (-OH) and ethylene oxide (EO) [95,97,221]. The successful formulation of highly CO_2_-permeable and selective membranes using polymers containing EO such as polyethylene glycol (PEG) and PEBAX [222,223,224] is the main aspiration for utilizing these chemical agents to functionalize nanofillers. The work by Li et al. [225] demonstrated the enhancement in permeability and CO_2_/N_2_ selectivity of the resulting nanocomposite from 250 barrer to 710 barrer and from 59 to 62, respectively, by functionalizing GO with PEG. A quadrupole–dipole interaction between CO_2_ and polar groups is the main mechanism that contributed to this improvement [196,226,227,228]. Similarly, a quadrupole–dipole interaction also contributes to the binding of CO_2_ with many metallic compounds from the transition group, especially zinc, cobalt, copper and titanium [229,230,231] and nanomaterials that are rich with electronegative groups such as silica, titanium dioxide (TiO_2_), GO and BN [32,124,208,232,233]. In addition, the formation of temporary sigma (σ) and pi (π) bonds between CO_2_ and metal ion has been proposed [234]. Functionalization of nanofillers using metallic compounds can be carried out via ion-substitution [206], doping [83] or co-synthesis [161]. The work by Ebadi Amooghin et al. [206], discussed in Section 3.2, is an example of nanofiller functionalization via ion-substitution by replacing the Na^+^ core of zeolite Y with Co^2+^. Meanwhile, doping and co-synthesis techniques are usually applied on non-metallic nanofillers. For instance, Ogieglo et al. [148] enhanced the sieving properties of their carbon molecular sieve (CMS) via doping with aluminum oxide (Al_2_O_3_). The authors reported a nearly 10-fold increment in CO_2_/CH_4_ and H_2_/N_2_ selectivity when the dopant content in CMS was increased through five vapor phase infiltration cycles. However, the permeation rate of gases was negatively affected due to partial pore blockage by Al_2_O_3_ which led to reduction in overall porosity of the CMS. In a separate study, Zhou et al. [235] successfully synthesized carbon nitride (CN)-TiO_2_ hybrid photocatalyst via co-synthesis technique by pyrolyzing mixed precursors of urea and titanium disulfate. The CN-TiO_2_ obtained from optimized precursor composition exhibited CO_2_ photoreduction efficiency, which was far superior compared to that of individual nanomaterials.

Another chemical which has gained traction in the field of CO_2_ separation is IL, a type of salt that exists as liquid at temperature below 100 °C [236]. The interest on IL is largely driven by its operational stability and large CO_2_ uptake [237]. IL, especially those consisting of imidazolium-based cations and fluorinated anions showed excellent CO_2_-affinity and can be processed into supported IL membranes [38,238] and poly(ionic liquid) membranes [239,240] which exhibit good separation performance [37,97,241,242]. The recent development of CO_2_ separative NCMs has focused on the use of IL, especially 1-butyl-3-methylimidazolium hexafluorophosphate ([Bmim][PF_6_]), 1-butyl-3-methylimidazolium tetrafluoroborate ([Bmim][BF_4_]) and [Bmim][Tf_2_N] as nanofiller functionalization agents. IL functionalization was usually achieved via surface coating or infiltration of IL into the pore of nanofillers, as demonstrated by the works of Rhyu et al. [97] and Chae et al. [243], respectively. Both studies employed [Bmim][BF_4_] as a functionalization agent and found that NCMs incorporated with the IL-functionalized nanomaterials exhibited significantly higher CO_2_ permeance compared to counterparts with pristine nanomaterials. This enhancement was ascribed to the presence of [BF_4_]^−^, which preferentially interacts with CO_2_, hence, facilitating the transport of CO_2_ across the NCMs.

The introduction of CO_2_-affinitive functional groups can lead to the creation of ‘reverse selective’ membrane whereby the permeation of CO_2_ (3.3 Å) is more favorable than that of smaller/lighter gases especially H_2_ (2.9 Å) [244,245,246]. Compared to H_2_-selective membrane, reverse selective membrane is deemed more practical for separation of H_2_-CO_2_ pair because CO_2_ is directly removed from the system while purified H_2_ which is still pressurized is retained and passed down the production line [36,247]. This could contribute to a substantial cost saving for the purification process as recompression of H_2_ stream is omitted.

As summarized in Table 1, the incorporation of nanofiller could boost CO_2_ permeability and selectivity of membranes in range of 10–1900% and 5–670%, respectively. On the other hand, NCMs containing FNs could perform 5–180% better than those with pristine nanofiller. Among the literature, the work by Liu et al. [82], who successfully developed high performing TFC with CO_2_ permeance upwards of 7000 GPU and CO_2_/N_2_ of 26 deserves particular attention. The architecture of their membrane comprises an ultrathin (650 nm) UiO-66-NH_2_-infiltrated PIM selective layer that was spun-coated on a porous PAN substrate that was pre-coated with a PDMS gutter layer containing amorphous MOF nanosheets. By virtue of the high FFV of its structure, PIM is an excellent candidate for the development of a high-performing gas separation membrane. As discussed in Section 2, the incorporation of MOF helps retain the structure of PIM by functioning as connector between adjacent polymer chains which promoted crosslinking of the polymeric network, thus giving NCM that is more resistive to aging. Additionally, attributed to the good nanofiller–polymer interaction which yielded defect-free NCM, the UiO-66-NH_2_-enhanced PIM was 73% more CO_2_ permeable and 37% more selective than pristine PIM. All these showcased the multi-effects feature of functionalized nanomaterials, including materializing formation of defect-free ultrathin nanocomposite layer, enhancing separation performance and extending the service-life of NCM.

**Table 1 membranes-12-00186-t001:** Characteristics of CO_2_-separative NCMs containing FN reported between year 2017 and 2021.

Base Polymer	Filler (Loading)	Modification	Test Conditions	$P_{{\mathrm{CO}}_{2}}$	${\bar{P}}_{{\mathrm{CO}}_{2}}$	$\Delta P_{{\mathrm{CO}}_{2}}$	$\alpha_{N_{2}}^{{\mathrm{CO}}_{2}}$	$\alpha_{{\mathrm{CH}}_{4}}^{{\mathrm{CO}}_{2}}$	$\Delta \alpha_{N_{2}}^{{\mathrm{CO}}_{2}}$	$\Delta \alpha_{{\mathrm{CH}}_{4}}^{{\mathrm{CO}}_{2}}$	Ref
PC	APTMS-SiO_2_ (3 wt.%)	co-condensation of APTMS with hydrolyzed TEOS (SiO_2_ precursor)	6 bar, 24 °C pure gas	-	20	340 *	38	29	98 *	57 *	[248]
crosslinked PEO	2-amBzIM-ZIF-7 (30 wt.%)	ligand substitution of BzIM by 2-amBzIM (70% substitution)	5 bar, 35 °C CO_2_:CH_4_ = 50:50 mol. ratio	213	-	10 *	-	56	-	167 *	[56]
PEBAX-1657	APTES-silica (15 wt.%)	grafting with acid hydrolyzed APTES	10 bar, 25 °C, pure gas	174	-	36 * 26 ^ϯ^	-	40.2	-	71 * 18 ^ϯ^	[32]
PEBAX-1657	UiO-66-NH_2_ (1.5 wt.%)	substitution of 1,4-dicarboxybenzene to 2-amino-1,4-dicarboxybenzene	7 bar, 25 °C pure gas	393	-	61 * 7 ^ϯ^		40		88 * 27 ^ϯ^	[111]
PEBAX-1657	UiO-66-NH_2_ (50 wt.%)	substitution with amine containing ligand	2 bar, 25 °C CO_2_:CH_4_ = 20:80 mol. ratioCO_2_:N_2_ = 20:80 mol. ratio	-	338	106 * 50 ^ϯ^	57	21	43 * 27 ^ϯ^	67 * 24 ^ϯ^	[126]
XTR-PI	Am-BN (1 wt.%)	ball-mill with urea	-bar, 25 °C pure gas	21	-	−89 *	-	69	-	212 *	[124]
CNF/MCE	UiO-66-NH_2_	carboxylation of CNF and replacing UiO-66 ligand with ATA	2 bar, 25 °C pure gas	139	-	1886 * 178 ^ϯ^	46	-	667 * 64 ^ϯ^	-	[204]
PEBAX-1657/PEG	TEPA-TNT (3 wt.%)	coating TNT with TEPA	5 bar, 35 °C pure gas	168	-	68 *	-	16	-	12 *	[90]
PIM	UiO-66-NH_2_ (10 wt.%)	amine containing ligand	1 bar, 35 °C pure gas	-	7460	73 *	26	-	37 *	-	[82]
PA	PEI-BN	coating with PEI	3 bar, 25 °C pure gas	-	47	37 * 5 ^ϯ^	47	-	20 * 21 ^ϯ^	-	[93]
PEBAX-1657/PVC	PEBAX/SiO_2_	priming with host matrix polymer	1 bar, 25 °C pure gas	-	29	63 *	76	-	36 *	-	[249]
PVA	MPEG-TiO_2_ (3 wt.%)	grafting of MPEG via radical polymerization	10 bar, 35 °C pure gas	5.4	-	476 *	49	6.1	31 *	26 *	[250]
PMMA	MPEG-TiO_2_ (5 wt.%)	grafting of MPEG via radical polymerization	10 bar, 35 °C pure gas	32	-	1081 *	57	4.2	55 *	19 *	[151]
PDMS	PEO-Si	nucleophilic addition of epoxy group of GOTMS (Si-precursor) with amine group of Jeffamine ED-2003	2 bar, 25 °C pure gas	-	3636	21 *	28.2	-	103 *	-	[95]
PSF	GOTMS-SiO_2_ (20 wt.%)	adsorption	10 bar, 30 °C pure gas	13	-	75 *	46	36	42 *	25 *	[251]
PEBAX-1074	OA-ZnO (8 wt.%)	esterification	2 bar, 25 °C pure gas	152	-	38 *	62	14	24 *	22 *	[84]
PEBEX-1074	OMWCNT (2.5 wt.%)	acid oxidation	2 bar, 25 °C pure gas	134	-	106 *	-	21	-	17 *	[252]
PMMA-co-MA-PEG/PC	OGNR (3 wt.%)	HNO_3_ treated GNR	0.7 bar, 27 °C pure gas	140	-	20 *	42	-	107 *	-	[125]
PIM	OH-pDCX (5 wt.%)	hydroxylation via Friedel-Crafts reaction	2 bar, 25 °C pure gas	8510	-	14 *	28	22	22 *	24 *	[135]
PMP	hydrolyzed TNT (2 wt.%)	treatment of TNT with strong base	2 bar, 25 °C pure gas	269	-	445 *	224	70	155 *	224 *	[232]
PEBAX-1074	[Bmim][PF_6_]/SiO_2_ (8 wt.%)	coating of IL on SiO_2_ (10:1 wt./wt.)	2 bar, 25 °C pure gas	154	-	47 *	-	19	-	3 *	[39]
PSF	[Bmim][BF_4_]@KIT-6 (100% selective layer)	immobilization of [Bmim][BF_4_] by KIT-6 (1:0.2 *w*/*w*)	2 bar, r.t., pure gas	-	51.6	204 *	5.4	4.8	7 *	0 *	[243]
6FDA-ODA	[Bmim][Tf_2_N]@UiO-66-PEI (15 wt.%)	post-synthetic modification with PEI and IL	1 bar, 35 °C CO_2_:CH_4_ = 50:50 mol. ratio	26	-	152 *	-	60	-	66 *	[99]
PEO	[Bmim][BF_4_]/ZnO (0.8 wt.%)	coating ZnO with [Bmim][BF_4_] (1:2 *w*/*w*)	r.t. pure gas	-	36	225 *	30	-	357 *	-	[97]
Matrimid-5218	[Co(tetra-aza)]^2+^-NaY (15wt.%)	ion exchange and complex formation	2 bar, 35 °C CO_2_:CH_4_ = 10:90 mol. ratio	19	-	127 * 8 ^ϯ^	-	112	-	207 * 158 ^ϯ^	[206]
PEBAX-1657	CuZnIF (0.5 wt.%)	inclusion of second metal to ZIF and PEBAX priming	6 bar, 30 °C pure gas	148	-	48 *	162	46	51 *	42 *	[83]
PEBAX-1657	PSS-HNT (0.1 wt%)	grafting	3 bar, 25 °C pure gas	-	10	74 * 0 ^ϯ^	245	-	457 * 32 ^ϯ^	-	[146]
PLA	LCNF (6.5 wt.%)	grafting of CNF	0.4 bar, 37 °C pure gas	0.6	-	58 *	21	-	22 *	-	[200]
PEBAX-1657/PES	Nf/TiO_2_ 0.075:3 filler:polymer wt. ratio	coating TiO_2_ with Nf (1:0.045 *w*/*w*)	2.5 bar, pure gas	${\bar{P}}_{{\mathrm{SO}}_{2}}$ = 1671			$\alpha_{N_{2}}^{{\mathrm{SO}}_{2}}$ = 2928				[253]
6FDA-TP	ZIF-90 (40 wt.%)	condensation polymerization of 6FDA with TP	9.8 bar, 35 °C pure gas	45	-	125 *	20	36	0 *	2.7 *	[254]
Polyactive	m-ZnTCPP (used as gutter layer)	Surfactant assisted synthesis in absence of pyrazine	3.5 bar, 35 °C CO_2_:N_2_ = 10:90 mol. ratio	-	2160	-	31	-	-	-	[210]
PVA/PSF	PCNF (1 wt.%)	phosphorylation of CNF with diammonium hydrogen phosphate	5 bar, CO_2_:CH_4_ = 40:60 mol. ratio	-	78	200 *	-	45	-	55 *	[37]

* = change in performance relative to neat polymeric membrane; ^ϯ^ = change in performance relative to membrane incorporated with base-filler (non-modified filler); ${\bar{P}}_{i}$ = gas permeance in GPU where ‘*i*’ refers to gas species; $P_{i}$ = gas permeability in barrer where ‘*i*’ refers to gas species; $\Delta P_{i}$ = percentage of change in permeability/permeance (%), +ve value = improvement, −ve value = deterioration, $\Delta P_{i}=\frac{{Χ}_{\mathrm{NCM}}{\;-\;Χ}_{\mathrm{neat}}}{{Χ}_{\mathrm{neat}}}\;\times 100\%$, where X_NCM_ is the $P_{i}{\;\mathrm{or}\;\bar{P}}_{i}\;$ of NCM while X_neat_ is the $P_{i}{\;\mathrm{or}\;\bar{P}}_{i}$ of neat polymeric membrane; $\alpha_{j}^{i}$ = selectivity of gas ‘*i*’ over gas ‘*j*’; $\Delta \alpha_{j}^{i}$ = percentage of change in selectivity (%), +ve value = improvement, −ve value = deterioration, $\Delta \alpha_{j}^{i}=\frac{\alpha_{j}^{i}{}_{\mathrm{NCM}}\;{-\;\alpha }_{j}^{i}{}_{\mathrm{neat}}}{\alpha_{j}^{i}{}_{\mathrm{neat}}}\;\times 100\%$, where $\alpha_{j}^{i}{}_{\mathrm{NCM}}$ is the $\alpha_{j}^{i}$ of NCM while $\alpha_{j}^{i}{}_{\mathrm{neat}}$ is the $\alpha_{j}^{i}$ of neat polymeric membrane; [Co(tetra-aza)]^2+^-NaY = cobalt complex with tetraaza macrocyclic ligand encapsulated within zeolite; 2-amBzIM = 2-aminobenzimidazole; 6FDA = 4,4′-(hexafluoroisopropylidene)diphthalic anhydride; 6FDA = 4,40-hexa- fluoroisopropylidine bisphthalic dianhydride; Am-BN = amino modified boron nitride; APTES = (3-aminopropyl) triethoxysilane; APTMS = 3-aminopropyl trimethoxysilane; ATA = aminoterephthalic acid; BF_4_ = tetrafluoroborate; Bmim = 1-butyl-3-methylimidazolium; BzIM = benzimidazole; CNF = cellulose nanofiber; co = copolymerized; CoTCPP = cobalt tetrakis(4-carboxyphenyl)porphyrin); CuBDC = copper 1,4-benzenedicarboxylate; CuZnIF = copper-zinc bimetallic imidazolate framework; GNR = graphene nanoribbon; GOTMS = (3-glycidoxypropyl) trimethoxysilane; GOTMS = 3-glycidyloxypropyltrimethoxysilane; Jeffamine ED-2003 = *O,O’*-bis(2-aminopropyl)polypropylene glycol-block-polyethylene glycol-block-polypropylene glycol; KIT-6 = bicontinuous cubic mesostructured silica with Ia3d symmetry and interpenetrating cylindrical pores [255]; LCNF = lauryl functionalized nanocellulose fiber; MA = methacrylic amide; MCE = mixed cellulose ester; MPEG = methoxy poly(ethylene glycol) methacrylate; MPS = 3-methacryloxypropyl-trimethoxysilane; m-ZnTCPP = modified zinc (II) tetrakis(4-carboxyphenyl)porphyrin); Nf = Nafion, sulfonated PTFE with perfluorinated-vinyl-polyether side chains; OA-ZnO = oleic acid modified zinc oxide; ODA = 4,4′- Oxidianiline; OGNR = oxidized graphene nanoribbon; OH-pDCX = hydroxylated poly-dichloroxylene; OMWCNT = oxidized MWCNT; PA = polyamide; PC = polycarbonate; PCNF = phosphorylated cellulose nanofiber; PDMS = polydimethylsiloxane; PEBAX-1074 = poly(ether-block-amide), copolymer with 55 wt.% PEO and 45 wt.% PA; PEBAX-1657 = poly(ether-block-amide) copolymer with 60 wt.% PEO and 40 wt.% PA; PEG = polyethylene glycol; PEI = polyethyleneimine; PEO = polyethylene oxide; PES = polyethersulfone; PF_6_ = hexafluorophosphate; PI = polyimide; PIM = polymer of intrinsic microporosity; PLA = poly(lactic acid); PMMA = poly(methyl methacrylate); PMP = poly(4-methyl-1-pentene); Polyactive = poly(ethylene oxide)/poly(butylene terephthalate) copolyether ester; PSF = polysulfone; PSS-HNT = poly(sodium-p-styrene sulfonate) grafted halloysite nanotube; PTFE = polytetrafluoroethylene; PVA = polyvinyl alcohol; PVC = polyvinyl chloride; PVDF = poly(vinylidene fluoride); r.t. = room temperature; SiO_2_ = silica; TEOS = tetraethyl orthosilicate; TEPA = tetraethylene pentaamine; Tf_2_N = bis(trifluoromethylsulfonyl)imide; TNT = titanium dioxide nanotube; TP = triptycene; UiO = University of Oslo MOF; wt. = weight; XTR = crosslinked thermally rearranged; ZIF = zeolitic imidazolate framework.

H_2_ and O_2_ are nonpolar gases that have limited interaction with polymeric materials [115,151]. Therefore, separation of these gases by polymeric membrane is usually dominated by a sieving mechanism. On the other hand, H_2_ is affinitive to group 10 metals, especially palladium (Pd), hence, metallic membrane or metal coated ceramic membrane are widely developed for H_2_ separation and showed good efficacy [85,129,256,257]. In the field of NCM, membranologists often exploit this H_2_–metal interaction to design nanomaterials that are selective toward H_2_. This can be seen from the compilation in Table 2, whereby Pd-functionalization via doping or coating was employed to enhance the characteristic of nanofillers. Yet, the study by Patel and Acharya [258] suggested that Pd could be deactivated if H_2_ was adsorbed at low temperature. As such, it is necessary to circumvent this problem by altering the crystal structure of Pd through doping with other metals. Metal doping could enhance the interaction with H_2_ and increase the availability of active sites which boosted the gas sorption capability of Pd. Meanwhile, based on Table 3, there is no specific functionalization agent that is used to target the separation of O_2_. This is understandable as the contemporary O_2_-affinitive materials, commonly known as mixed ionic-conductive (MIC) materials, are usually operated at temperatures above 500°C to enable solvation of the gas into the MIC [47,259,260]. This limits the development of MIC-based nanofiller that can be used effectively alongside polymers. Based on the data compiled in Table 1, Table 2 and Table 3, the incorporation of FN can boost membrane performance in a range of 3–2000% depending on the type of nanofiller employed and the characteristics of the pristine polymeric matrix.

#### 3.3.2. Nanofiller Hybridization

The synergy of multiple nanofillers especially among those with different dimensions can bring about benefits similar to that of functionalization, including enhancing nanofiller dispersion, promoting adhesion with polymer matrix and improving gas selectiveness [164,261,262,263]. Wong et al. [81] and Zhang et al. [264] demonstrated that hybridizing CNT with GO could improve the overall stability of the suspension in aqueous solution. The myriad -OH groups at the edge of GO improved the hydrophilicity of the hybrid whereas CNT was anchored to the graphitic base of GO via hydrophobic interactions. The intercalation of CNT in-between GO diminished entanglement of the nanotubes and prevented restacking of the nanosheets. The difference in dispersity between nanotubes suspension and nanosheet-nanotube suspension is clearly shown in Figure 6a,b. Mirzae et al. [85] were able to elevate the H_2_ permeability and H_2_/N_2_ selectivity of their nanocomposites by 54% and 56%, respectively, by inserting Pd nanoparticles into the core of ZIF-8 nanofiller (Figure 6c). In a different work, by depositing CNC onto reduced GO (rGO), You et al. [162] converted their porous nanosheet (Figure 6d) into perfect gas barrier (Figure 6e). This modification allowed the authors to fabricate nanocomposite with enhanced tortuosity (Figure 6g), which exhibits elevated resistance toward diffusion of O_2_ and water vapor. All these examples illustrated the tremendous potential of nanofiller hybridization for creating new advanced materials that could expand the development of gas-separative NCMs. The data of some recent studies that adopted a nanofiller-hybridization approach to enhance the membrane performance are tabulated in Table 4. From the compiled data, hybridization of different nanofillers could elevate the overall gas permeability and selectivity of the resulting nanocomposites by 10–800% and 10–125%, respectively, compared to their counterparts containing single filler.

## 4. Future Prospects

A comparison between the separation data of NCMs containing FN developed in the last 5 years to the Robeson’s curve revised in 2008 provides a glimpse into the progress of polymeric membrane gas separation technologies over the past decade. From Figure 7, it can be clearly seen that a large number of CO_2_-separative NCMs containing FN exhibited separation performance that outclass most of the polymeric membranes from before 2008, whereas those designed for H_2_ and O_2_ separation did not celebrate the same success. This can be ascribed to the lack of chemical agents with affinity to H_2_ and O_2_ when operated close to ambient temperature. The low number of publications related to the separation of these two gases using NCMs containing FN over the past 5 years is a strong indication of the lack of enthusiasm in advancing this technology for the field of H_2_ and O_2_ separation. In fact, instead of separation application, chemically modified nanomaterials are currently heavily invested in for applications that can generate H_2_ and O_2_, especially water splitting, as these gases are important precursors for green energy technologies. In terms of CO_2_-separative NCMs, amine-functionalization showed the greatest number of successes, whereby the membranes from 5 out of 10 works (or 50%) exhibited performance above Robeson’s boundary. Future works aiming to up-scale NCM for CO_2_ separation can take hint from these encouraging results to develop a high-performance membrane by adopting amine-functionalization to enhance their nanofillers. Other functionalization that deserves attention includes nanofiller hybridization (33% success), hydroxylation (40% success) and metal doping (50% success). Considering IL as a relative new class of material in the space of CO_2_ separation, more research efforts can be placed on IL-functionalized nanomaterial. The work by Carvalho et al. [266] in 2010 showed that phosphonium-based IL could exhibit CO_2_-sorption capacity greater than that of imidazolium-based IL. A decade later, Voskian et al. [267] reported the development of amine-functionalized IL which, instead of relying on thermal-based processes, can be regenerated electrochemically. Recent studies by Liu et al. [268,269] found that the regeneration of their mixed solvents system, which comprises amino-functionalized IL, consumed 40% less energy than aqueous amine solution, while exhibiting 3.56 times higher CO_2_ absorption loading at 1.78 mol·mol^−1^. These studies depicted the potential of IL-based CO_2_ capture technologies which are still being vigorously developed. In fact, the variety of cations and anions currently available (and still expanding) is so vast that advanced computer simulation and machine learning could play significant roles to screen, identify and optimize IL structures [237,270,271,272].

Analysis of the permeability in Robeson’s plot provides information about the intrinsic separation characteristic of the materials but does not represent the entirety of the membrane. The fabrication technique employed also plays a crucial part in the performance of the final product. Many researchers recognized the need for a membrane with an ultrathin selective layer to achieve high productivity, which can be produced via advanced fabrication techniques such as spin coating, interfacial polymerization, layer-by-layer and chemical deposition [11,273,274,275]. Recent works explored the use of nanomaterials as interlayer or gutter layer to control the thickness of the selective layer [121,210]. For example, the work by Ji et al. [95] demonstrated that the deposition of copper hydroxide nanofiber (CHN) as an interlayer could fine-tune the pore size of the PAN support layer while retaining high porosity. This prevented the penetration of PDMS coating, which led to the formation of an ultrathin selective layer (≈100 nm) that was 10 times thinner and 2.5 times more permeable than those without a CHN interlayer. The resulting nanocomposite exhibited excellent CO_2_ permeance close to 3000 GPU with CO_2_/N_2_ selectivity of 28. Similarly, Wang et al. [276] were able to reduce the thickness of an interfacially polymerized selective layer from 200 nm to 120 nm by modifying the PES support layer with a hydrophilic covalent organic framework (COF) interlayer. Apart from the formation of an ultrathin selective layer, some work also showed promising separation performance by controlling the orientation of incorporated nanofillers via electrical- or shear-assisted alignment methods [164,277]. All these exemplify the importance of the fabrication technique in the development of high performing membranes. Nevertheless, functionalization will continue to reinforce the implementation of new techniques devised for nanocomposite fabrication. For instance, the attachment of a nanofiller interlayer on a polymeric support can be improved via the introduction of adhesive bonding agents, or the response of nanofiller towards the electromagnetic field can be enhanced via introduction of ferrous components. Furthermore, the ability of chemical modification to alter the structure of nanofiller was proven useful by Liu et al. [210] to ensure the viability of their interlayer-mediated thin film formation. Using a surfactant-assisted approach, they inhibited the growth of MOF crystals, which resulted in the formation of MOF nanosheets that can be feasibly assembled into an ultrathin interlayer [57]. The flexibility of the nanosheet structure also hampers the formation of defects such as cracks that arise from bending stress, which in turn improves the stability and mechanical properties of the nanocomposite membrane.

As discussed, contemporarily, there are no standard methods to analyze and assess the effectiveness of functionalization on a nanofiller–polymer interaction. This makes comparing the modification agents difficult, especially among those applied on different nanofiller and polymer combinations. Additionally, the influences of post-incorporation processing on the properties of FNs and nanocomposites were not systematically studied. For example, the application of heat is known to reduce GO, which leads to the removal of polar groups [124], while the viscosity of IL would increase after CO_2_ sorption [278]. Yet, the effects of these changes on nanofiller chemical and structural properties were often overlooked. Essentially, the alteration and sensitivity of FNs towards environmental and post-processing changes were not well-documented. These are some areas that deserve attention from future research on FN-incorporated NCMs.

## 5. Conclusions

All in all, functionalization is an indispensable tool in the development of high-performance NCM. The introduction of chemical functional groups could enhance the interaction between the inorganic nanofiller and the organic polymer matrix, which promote defect-free NCM formation. Furthermore, functionalization enables tuning of the nanofillers separation characteristics. NCMs incorporated with FN often exhibit exceptional separation performance that outperform the neat counterpart. Despite the numerous benefits ascribed to functionalization, in-depth analysis and standardized characterization are required to streamline and fortify understanding on nanofiller–polymer interactions. On top of that, the impacts of post-nanofiller incorporation processes and operation conditions on FNs should not be overlooked.

## Figures and Tables

**Figure 1 membranes-12-00186-f001:**
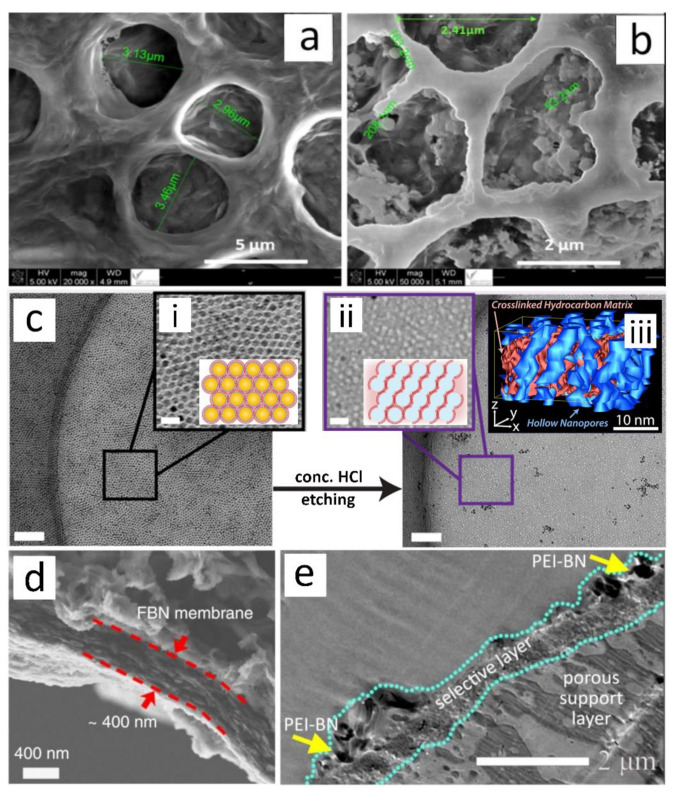
Cross sectional morphologies of (**a**) porous PVDF-HFP, (**b**) PEDA-SiO_2_ embedded PVDF-HFP [91], (**d**) thin film layer from assembled amino functionalized boron nitride (Am-BN) [92] and (**e**) © polyamide (PA) selective layer incorporated with polyethyleneimine functionalized boron nitride (PEI-BN) [93]. (**c**) Surface morphology of nanoparticle-templated polymer layer fabricated by chemically etching (**i**) crosslinked iron oxide (Fe_3_O_4_) to obtain (**ii**) ultrathin porous polymeric layer with (**iii**) vertically continuous pores [94].

**Figure 2 membranes-12-00186-f002:**
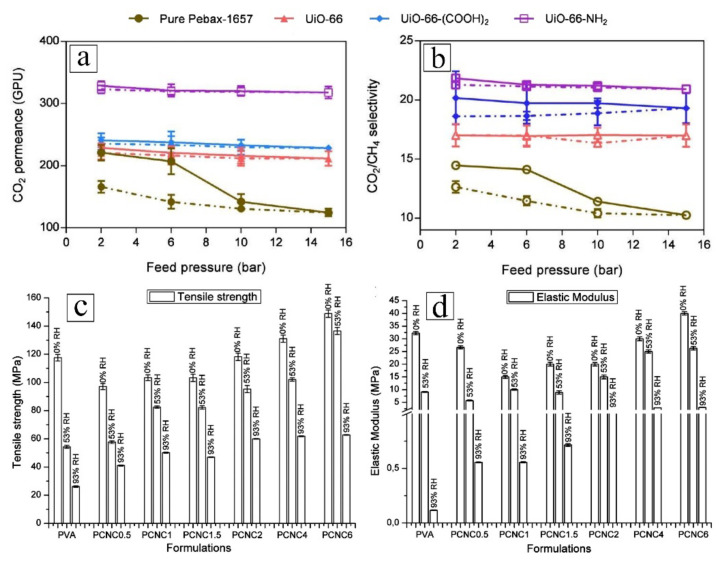
Trend of change in (**a**) CO_2_ permeance and (**b**) CO_2_/CH_4_ selectivity of neat PEBAX and nanocomposites containing pristine, carboxylated (UiO-66-(COOH)_2_) and aminated UiO-66 (UiO-66-NH_2_) during pressurization (solid line) and depressurization (dash line) [126]. (**c**) Tensile strength and (**d**) elastic modulus of PVA and PVA-based nanocomposites containing different CNC loading (PCNx, where x refers to CNC loading of 0.5–6 wt.%) at various moisture level [127].

**Figure 3 membranes-12-00186-f003:**
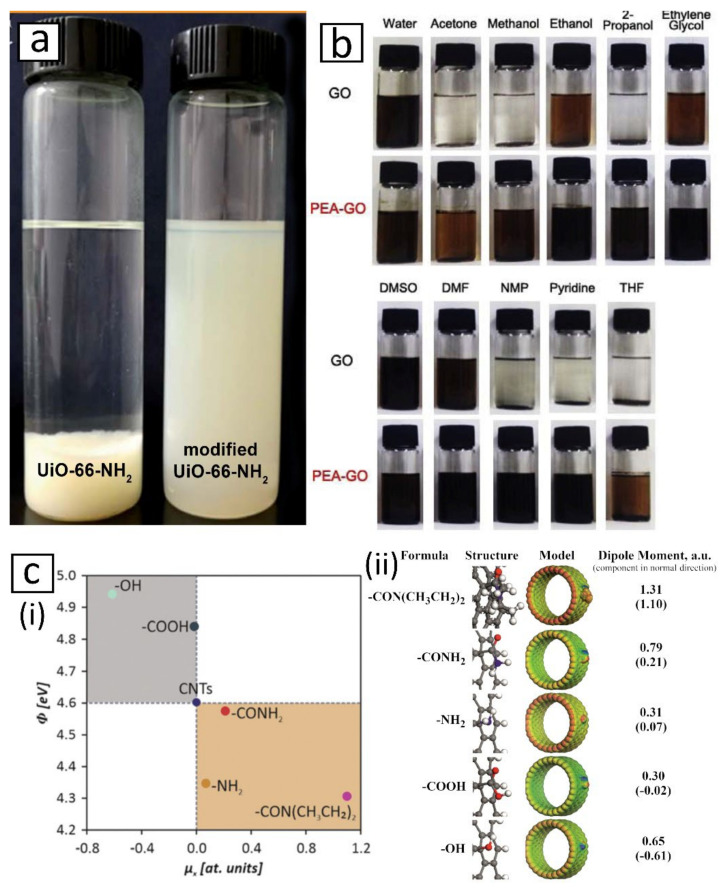
Suspensions of (**a**) UiO-66-NH_2_ and palmitoyl chloride modified UiO-66-NH_2_ in cyclohexane after 12 h standing [182] and (**b**) GO and PEA modified GO in various solvents after 50 days standing [173]. (**c**) (**i**) Plot of functionalized CNTs’ work function as a function of surface dipole moments and (**ii**) model corresponded to CNT surface functional groups [183].

**Figure 4 membranes-12-00186-f004:**
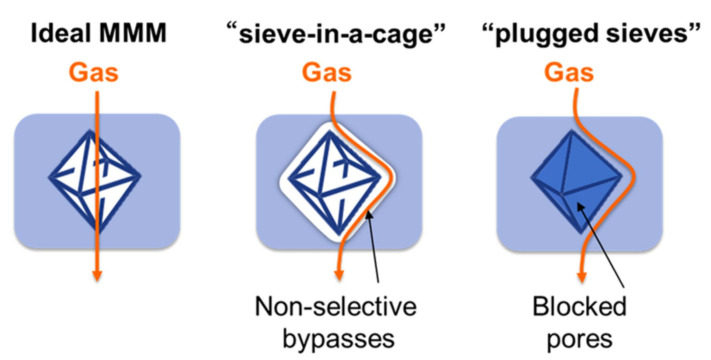
Ideal and undesirable nanofiller–polymer interface [184].

**Figure 5 membranes-12-00186-f005:**
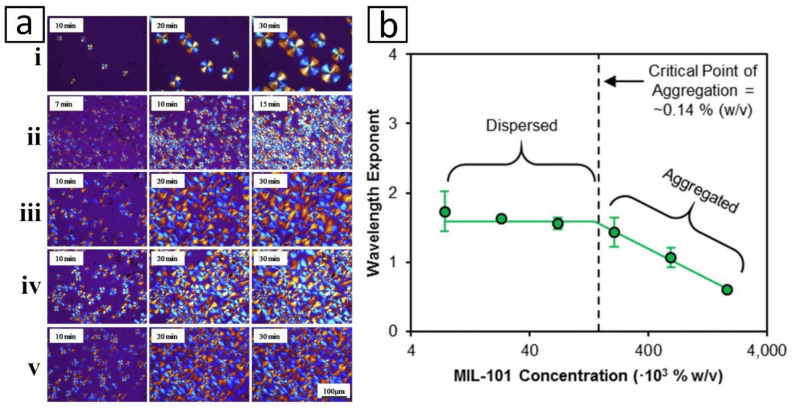
(**a**) Polarized optical microscopy images of crystallizing (**i**) PLA and PLA-based nanocomposite containing (**ii**) CNT, (**iii**) carboxyl CNT, (**iv**) hydroxyl CNT and (**v**) fluorinated CNT at 135 °C [185]. (**b**) Influence of chromium terephthalate MOF (MIL-101) loading on its dispersibility in nitrobenzene [193].

**Figure 6 membranes-12-00186-f006:**
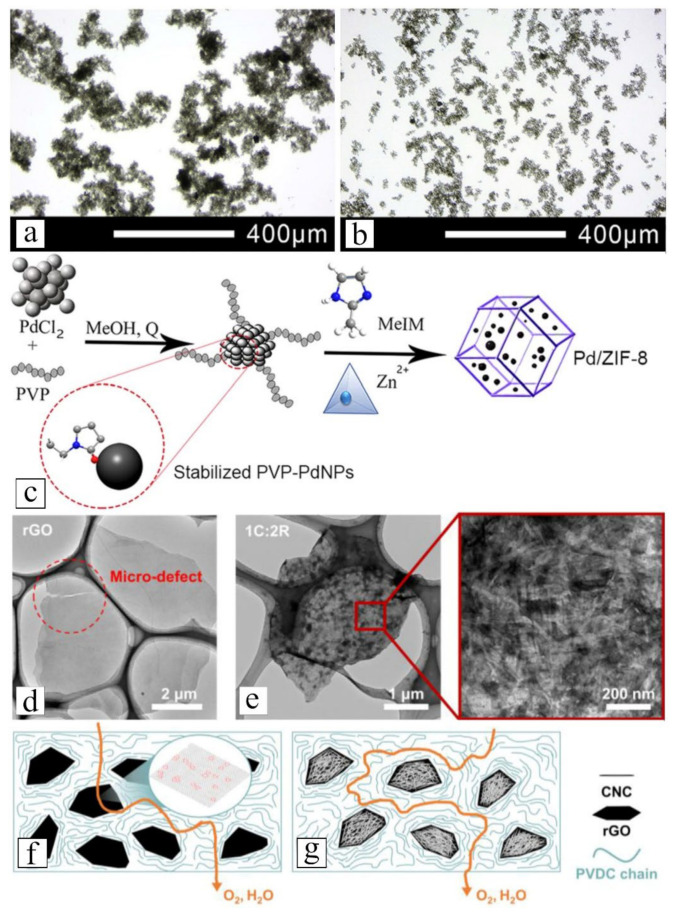
Optical microscopic images of (**a**) aminated CNT (ACNT) and (**b**) ACNT-GO dispersion [81], (**c**) illustration of encapsulation of Pd nanoparticle in ZIF-8 [85], TEM images of (**d**) rGO with micro-defect and (**e**) CNC-rGO as well as graphical illustrations of (**f**,**g**) molecular diffusion path across membrane matrixes that are embedded with rGO or CNC-rGO [162].

**Figure 7 membranes-12-00186-f007:**
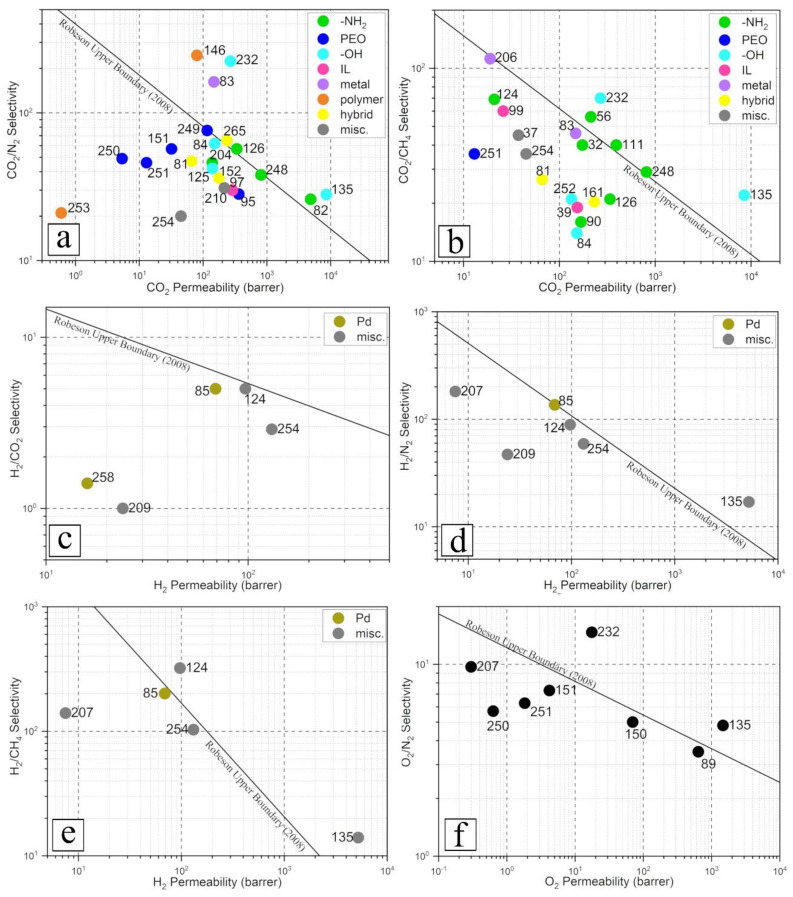
Overlaps of separation data from references on Robeson’s plots for (**a**) CO_2_/N_2_, (**b**) CO_2_/CH_4_, (**c**) H_2_/CO_2_, (**d**) H_2_/N_2_, (**e**) H_2_/CH_4_ and (**f**) O_2_/N_2_ gas pairs. The number beside each dot referred to the citation number of the corresponding reference.

**Table 2 membranes-12-00186-t002:** Characteristics of H_2_-separative NCMs containing FN reported between year 2017 and 2021.

Base Polymer	Filler (Loading)	Modification	Test Conditions	$P_{H_{2}}$	$\Delta P_{H_{2}}$	$\alpha_{{\mathrm{CO}}_{2}}^{H_{2}}$	$\alpha_{N_{2}}^{H_{2}}$	$\alpha_{{\mathrm{CH}}_{4}}^{H_{2}}$	$\Delta \alpha_{{\mathrm{CO}}_{2}}^{H_{2}}$	$\Delta \alpha_{N_{2}}^{H_{2}}$	$\Delta \alpha_{{\mathrm{CH}}_{4}}^{H_{2}}$	Ref
PC	Pt-Pd (surface coating)	metal dope	2 bar, 35 °C pure gas	16	21 *	1.4	-	-	−8 *	-	-	[258]
Matrimid	Pd@ZIF-8 (20 wt.%)	encapsulation of Pd in ZIF-8 cage	5 bar, 25 °C pure gas	69	140 * 54 ^ϯ^	5	136	201	73 * 52 ^ϯ^	47 * 56 ^ϯ^	62 * 59 ^ϯ^	[85]
6FDA-TP	ZIF-90 (40 wt.%)	condensation polymerization of 6FDA with TP	9.8 bar, 35 °C pure gas	131	122 *	2.9	59	103	−3.3 *	−1.7 *	−8.8 *	[254]
PE	GOTMS-SiO_2_ (10 wt.%)	GOTMS as coupling agent	4 bar, 25 °C pure gas	24	23 *	1	47	-	−5 *	−11 *	-	[209]
XTR-PI	Am-BN (1 wt.%)	ball-mill with urea	-bar, 25 °C pure gas	97	−56 *	5	89	322	364 *	365 *	1210 *	[124]
PI	MPS-TiO_2_ (20 wt.%)	grafting of TiO_2_	3.5 bar, 35 °C pure gas	7.5	102 *	-	181	140	-	3 *	−42 *	[207]
PIM	OH-pDCX (5 wt.%)	hydroxylation via Friedel-Crafts reaction	2 bar, 25 °C pure gas	5230	16 *	-	17	14	-	25 *	27 *	[135]

* = change in performance relative to neat polymeric membrane; ^ϯ^ = change in performance relative to membrane incorporated with base-filler (non-modified filler); ${\bar{P}}_{i}$ = gas permeance in GPU where ‘*i*’ refers to gas species; $P_{i}$ = gas permeability in barrer where ‘*i*’ refers to gas species; $\Delta P_{i}$ = percentage of change in permeability/permeance (%), +ve value = improvement, −ve value = deterioration, $\Delta P_{i}=\frac{{Χ}_{\mathrm{NCM}}{\;-\;Χ}_{\mathrm{neat}}}{{Χ}_{\mathrm{neat}}}\;\times 100\%$, where X_NCM_ is the $P_{i}{\;\mathrm{or}\;\bar{P}}_{i}\;$ of NCM while X_neat_ is the $P_{i}{\;\mathrm{or}\;\bar{P}}_{i}$ of neat polymeric membrane; $\alpha_{j}^{i}$ = selectivity of gas ‘*i*’ over gas ‘*j*’; $\Delta \alpha_{j}^{i}$ = percentage of change in selectivity (%), +ve value = improvement, −ve value = deterioration, $\Delta \alpha_{j}^{i}=\frac{\alpha_{j}^{i}{}_{\mathrm{NCM}}\;{-\;\alpha }_{j}^{i}{}_{\mathrm{neat}}}{\alpha_{j}^{i}{}_{\mathrm{neat}}}\;\times 100\%$, where $\alpha_{j}^{i}{}_{\mathrm{NCM}}$ is the $\alpha_{j}^{i}$ of NCM while $\alpha_{j}^{i}{}_{\mathrm{neat}}$ is the $\alpha_{j}^{i}$ of neat polymeric membrane; Am-BN = amino functionalized boron nitride; GOTMS = g-glycidyloxypropyltrimethoxysilane; PC = polycarbonate; Pd = palladium; PE = polyester; Pt-Pd = platinum doped palladium; SiO_2_ = silica; XTR-PI = crosslinked thermally rearranged polyimide.

**Table 3 membranes-12-00186-t003:** Characteristics of O_2_-separative NCMs containing FN reported between year 2017 and 2021.

Base Polymer	Filler (Loading)	Modification	Test Conditions	$P_{O_{2}}$	$\Delta P_{O_{2}}$	$\alpha_{N_{2}}^{O_{2}}$	$\Delta \alpha_{N_{2}}^{O_{2}}$	Ref
PDMS	PDMS-SiO_2_ (10 wt.%)	priming SiO_2_ with host polymer	2 bar, r.t. pure gas	640	8 *	3.5	42 *	[89]
PIM	OH-pDCX (5 wt.%)	hydroxylation via Friedel-Crafts reaction	2 bar, 25 °C pure gas	1470	13 *	4.8	20 *	[135]
PSF	GOTMS-SiO_2_ (20 wt.%)	adsorption	10 bar, 30 °C pure gas	1.8	43 *	6.3	17 *	[251]
PVA	MPEG-TiO_2_ (3 wt.%)	grafting of MPEG via radical polymerization	10 bar, 35 °C pure gas	0.63	2000 *	5.7	72 *	[250]
PMMA	MPEG-TiO_2_ (5 wt.%)	grafting of MPEG via radical polymerization	10 bar, 35 °C pure gas	4.2	1508 *	7.3	98 *	[151]
PMP	hydrolyzed TNT (2 wt.%)	treatment of TNT with strong base	2 bar, 25 °C pure gas	17.6	418 *	14.7	143 *	[232]
PI	MPS-TiO_2_ (20 wt.%)	grafting of TiO_2_	3.5 bar, 35 °C pure gas	0.3	81 *	9.7	−7.6 *	[207]

* = change in performance relative to neat polymeric membrane; ${\bar{P}}_{i}$ = gas permeance in GPU where ‘*i*’ refers to gas species; $P_{i}$ = gas permeability in barrer where ‘*i*’ refers to gas species; $\Delta P_{i}$ = percentage of change in permeability/permeance (%), +ve value = improvement, −ve value = deterioration, $\Delta P_{i}=\frac{{Χ}_{\mathrm{NCM}}{\;-\;Χ}_{\mathrm{neat}}}{{Χ}_{\mathrm{neat}}}\;\times 100\%$, where X_NCM_ is the $P_{i}{\;\mathrm{or}\;\bar{P}}_{i}\;$ of NCM while X_neat_ is the $P_{i}{\;\mathrm{or}\;\bar{P}}_{i}$ of neat polymeric membrane; $\alpha_{j}^{i}$ = selectivity of gas ‘*i*’ over gas ‘*j*’; $\Delta \alpha_{j}^{i}$ = percentage of change in selectivity (%), +ve value = improvement, −ve value = deterioration, $\Delta \alpha_{j}^{i}=\frac{\alpha_{j}^{i}{}_{\mathrm{NCM}}\;{-\;\alpha }_{j}^{i}{}_{\mathrm{neat}}}{\alpha_{j}^{i}{}_{\mathrm{neat}}}\;\times 100\%$, where $\alpha_{j}^{i}{}_{\mathrm{NCM}}$ is the $\alpha_{j}^{i}$ of NCM while $\alpha_{j}^{i}{}_{\mathrm{neat}}$ is the $\alpha_{j}^{i}$ of neat polymeric membrane.

**Table 4 membranes-12-00186-t004:** Compilation of gas separation nanocomposite consisting hybridized nanofiller reported between year 2017 and 2021.

Base Polymer (Filler Loading)	Base Filler	Secondary Filler (Method)	Test Conditions	$P_{{\mathrm{CO}}_{2}}$	$\Delta P_{{\mathrm{CO}}_{2}}$	$\alpha_{N_{2}}^{{\mathrm{CO}}_{2}}$	$\alpha_{{\mathrm{CH}}_{4}}^{{\mathrm{CO}}_{2}}$	$\Delta \alpha_{N_{2}}^{{\mathrm{CO}}_{2}}$	$\Delta \alpha_{{\mathrm{CH}}_{4}}^{{\mathrm{CO}}_{2}}$	Ref
PHS/PPS (10 wt.%)	zeolite	CNT (mixing under reflux)	0.68 bar, 27 °C pure gas	177	10 ^ϯ^	36.1	-	10 ^ϯ^	-	[152]
Matrimid-5218 (20 wt.%)	ZIF-8	GO (in situ ZIF-8 growth with GO)	1 bar, 30 °C pure gas	238	358 * 34 ^ϯ^	65	-	80 * 55 ^ϯ^		[265]
PMP/PEBAX-1657 (3 wt.%)	carboxylated CNF	UiO-66-NH_2_ (in situ growth)	6 bar, 25 °C CO_2_:CH_4_ = 50:50 vol. ratio	232	31 ^ϯ^	-	20	-	93 ^ϯ^	[161]
PA (0.25 mg/mL)	ACNT	GO (mixing)	6 bar, 30 °C pure gas	66.3	23 * 20 ^ϯ^	47.1	26.5	39 * 23 ^ϯ^	35 * 19 ^ϯ^	[81]
Base Polymer (filler loading)	Base Filler	Secondary Filler (method)	Test Conditions	$P_{O_{2}}$	$\Delta P_{O_{2}}$	$F_{O_{2}}$	$\Delta F_{i}$	$\alpha_{N_{2}}^{O_{2}}$	$\Delta \alpha_{N_{2}}^{O_{2}}$	Ref
PSF (0.1 wt.%)	GO-NH_4_^+^	mSiO_2_	1 bar, 5–50 °C pure gas	70	1406 * 801 ^ϯ^			5	115 * 33 ^ϯ^	[150]
PSF	MWCNT	rGO	1 bar, 15–45 °C pure gas			722	45 ^ϯ^	2.4	34 ^ϯ^	[164]
Base Polymer (filler loading)	Base Filler	Secondary Filler (method)	Test Conditions	$F_{{\mathrm{SO}}_{2}}$	$\Delta F_{{\mathrm{SO}}_{2}}$		$R_{{\mathrm{SO}}_{2}}$	$\Delta R_{{\mathrm{SO}}_{2}}$		Ref
PVDF (40 wt.%)	zeolite 4A	Cu nanosheet (growth of Cu shell on zeolite core)	1 bar, 25 °C feed gas: SO_2_ liquid sorbent: NaOH (30 L/h)	9 × 10^−4^	107 *		74	124 *		[96]

* = change in performance relative to neat polymeric membrane; ^ϯ^ = change in performance relative to membrane incorporated with base-filler (non-modified filler); ${\bar{P}}_{i}$ = gas permeance in GPU where ‘*i*’ refers to gas species; $P_{i}$ = gas permeability in barrer where ‘*i*’ refers to gas species; $\Delta P_{i}$ = percentage of change in permeability/permeance (%), +ve value = improvement, −ve value = deterioration, $\Delta P_{i}=\frac{{Χ}_{\mathrm{NCM}}{\;-\;Χ}_{\mathrm{neat}}}{{Χ}_{\mathrm{neat}}}\;\times 100\%$, where X_NCM_ is the $P_{i}{\;\mathrm{or}\;\bar{P}}_{i}\;$ of NCM while X_neat_ is the $P_{i}{\;\mathrm{or}\;\bar{P}}_{i}$ of neat polymeric membrane; $\alpha_{j}^{i}$ = selectivity of gas ‘*i*’ over gas ‘*j*’; $\Delta \alpha_{j}^{i}$ = percentage of change in selectivity (%), +ve value = improvement, −ve value = deterioration, $\Delta \alpha_{j}^{i}=\frac{\alpha_{j}^{i}{}_{\mathrm{NCM}}\;{-\;\alpha }_{j}^{i}{}_{\mathrm{neat}}}{\alpha_{j}^{i}{}_{\mathrm{neat}}}\;\times 100\%$, where $\alpha_{j}^{i}{}_{\mathrm{NCM}}$ is the $\alpha_{j}^{i}$ of NCM while $\alpha_{j}^{i}{}_{\mathrm{neat}}$ is the $\alpha_{j}^{i}$ of neat polymeric membrane; $F_{O_{2}}$ = O_2_ permeation flux in cc·m^−2^·day^−1^; $F_{{\mathrm{SO}}_{2}}$ = SO_2_ absorption flux in mol·m^−2^·s^−1^; $\Delta F_{i}$ = percentage of change in flux (%), +ve value = improvement, −ve value = deterioration, $\Delta F_{i}=\frac{F_{i}{}_{\mathrm{NCM}}\;{-\;F}_{i}{}_{\mathrm{neat}}}{F_{i}{}_{\mathrm{neat}}}\;\times 100\%$, where $F_{i}{}_{\mathrm{NCM}}$ is the $F_{i}$ of NCM while $F_{i}{}_{\mathrm{neat}}$ is the $F_{i}$ of neat polymeric membrane; $R_{{\mathrm{SO}}_{2}}$ = percentage of SO_2_ removal efficiency; $\Delta R_{{\mathrm{SO}}_{2}}$ = percentage of change in removal efficience (%), +ve value = improvement, −ve value = deterioration; CNF = carbon nanofiber; GO-NH_4_^+^ = ammonium activated GO; mSiO_2_ = aminopropyl triethoxysilane (APTES) modified SiO_2_; PHS = poly(1-hexadecene-sulfone); PMP = polymethylpentyne; PPS = poly(1,4-phenylene sulfide).

## Data Availability

Not applicable.

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
