# Peer review of "The State-of-the-Art Functionalized Nanomaterials for Carbon Dioxide Separation Membrane"

_membranes, 2022, doi:10.3390/membranes12020186_

Round 1
Reviewer 1 Report
1. Plugged sieve is not discussed in text, suggest to explain it.
2. Line 414, incorporation of?
3. Suggest to briefly explain how to calculate change in performance or selectivity. Selectivity here refers to permselectivity or separation factor?
4. Please explain more for line 733 to 735.
5. Check grammar/writing for line 435, 437, 728, 791.
6. Suggest to write year beside the name for the citation.
Author Response
Thank you for your comments. Below are our point-by-point response:
Comment 1:
Plugged sieve is not discussed in text, suggest to explain it.
Response:
Thank you for the comments. Plugged sieve phenomenon has been added in Line to 504-509.
Comment 2:
Line 414, incorporation of?
Response:
The phrase have been revised to “nanofiller incorporation”
Comment 3:
Suggest to briefly explain how to calculate change in performance or selectivity. Selectivity here refers to permselectivity or separation factor?
Calculation of change in performance and selectivity have been added. Selectivity refers to permselectivity for studies using pure gas and separation factor for studies using mixed gas.
Comment 4:
Please explain more for line 733 to 735.
Response:
More explanation has been added for Line 733-735 (now Line 745-749)
Comment 5:
Check grammar/writing for line 435, 437, 728, 791.
Response:
Changed “interaction” to “interact” for Line 435 (now Line 439)
Changed “undergoes” to “undergo” for Line 437 (now Line 441)
Added “.” before “In the” for Line 728 (now Line 740)
Changed “enhance” to “enhanced” for Line 791 (now Line 811)
Comment 6:
Suggest to write year beside the name for the citation.
Response:
It is uncommon to write year beside the name for citation using numbered format. Authors think it is best to retain the current in-text citation format.
Reviewer 2 Report
Composite membrane is intensively investigated for gas separation. The recent progress on carbon dioxide separation was well summarized. To further improve the quality of this manuscript, several questions should be addressed.
- It might be more accuracy to emphasize CO2 separation rather than gas separation in the title. Otherwise, the authors are encouraged to survey more literatures on membrane separation of H2/CO2, H2/N2, H2/CH4 and O2/N2, which is very limited in Figure 7.
- How to define nanocomposite membrane? What is the different to the term of “mixed matrix membrane” and “composite membrane”?
- Line 90, it is not fear to emphasize “nano-sized inorganic material”. As I known, lots of literatures involving the organic filler have been published. Similar case is applied to line 116, wherein metal-organic framework should be included.
- Line 159-163: Figure 1a cannot support the statement of “a decrease in membrane pore size from 10.8 nm to 6.3 nm”. Please check the cited reference to confirm no information was missed.
- Please state the zeolite topology in Table 4.
Author Response
Thank you for the comments. Below are our point-by-point response
Comment 1:
It might be more accuracy to emphasize CO2 separation rather than gas separation in the title. Otherwise, the authors are encouraged to survey more literatures on membrane separation of H2/CO2, H2/N2, H2/CH4 and O2/N2, which is very limited in Figure 7.
Response:
The title has been revised accordingly
Comment 2:
How to define nanocomposite membrane? What is the different to the term of “mixed matrix membrane” and “composite membrane”?
Response:
Polysulfone (PSF) can be formed into asymmetric or dense membrane (both having single distinct layer) but when PSF was formed on a polyethylene terephthalate (PET) backing (PSF/PET), a composite membrane was obtained (PSF/PET has two distinct layers). Similarly, incorporation of carbon nanotube into PSF gives a single distinct piece of MMM (CNT@PSF) but if this layer is form on PET, a thin film nanocomposite (CNT@PSF/PET) is resulted.
Comment 3:
Line 90, it is not fear to emphasize “nano-sized inorganic material”. As I known, lots of literatures involving the organic filler have been published. Similar case is applied to line 116, wherein metal-organic framework should be included.
Response:
The description in Line 90, has been revised to “nano-sized material” and metal organic framework has been included in Line 116 (now Line 120-121).
Comment 4:
Line 159-163: Figure 1a cannot support the statement of “a decrease in membrane pore size from 10.8 nm to 6.3 nm”. Please check the cited reference to confirm no information was missed.
Response:
We have checked the cited reference. The description is correct. Figure 1a depicts the morphology of PVDF-HFP which is porous while Figure 1b depicts the morphology of PVDF-HFP which is incorporated with PEDA-SiO2. From both figures we can observed that the cavities in PVDF-HFP network which are 3 µm in diameter decreased to roughly 2 µm when nanofillers were incorporated. Different analysis technique will generally yield different pore size values. From the cited reference, the average pore diameter obtain from BET analysis is a magnitude smaller than that perceived from the SEM images due to limitation in SEM analysis and the way the morphologies were presented. When inspecting the images carefully, fine pores can be observed from the walls of the cavities (yellow circles in the attached word file) which could be associated to the small pores estimated by BET.
Comment 5:
Please state the zeolite topology in Table 4.
Response:
The topology of the zeolite employed by ref 96 has been added but the topology of zeolite used by ref 152 was not specified in the paper.
